# Host factor prioritization for pan-viral genetic perturbation screens using random intercept models and network propagation

Simon Dirmeier[1,2], Christopher Dächert[3,4], Martijn van Hemert[5], Ali Tas[5], Natacha S. Ogando[5], Frank van Kuppeveld[6], Ralf Bartenschlager[7,8], Lars Kaderali[9], Marco Binder[3], Niko Beerenwinkel[1,2]*

1 Department of Biosystems Science and Engineering, ETH Zurich, Basel, Switzerland, 2 SIB Swiss Institute of Bioinformatics, Basel, Switzerland, 3 Research Group "Dynamics of Early Viral Infection and the Innate Antiviral Response" (division F170), German Cancer Research Center, Heidelberg, Germany, 4 Faculty of Biosciences, Heidelberg University, Heidelberg, Germany, 5 Department of Medical Microbiology, Leiden University Medical Center, Leiden, The Netherlands, 6 Virology Division, Department of Infectious Diseases and Immunology, Faculty of Veterinary Medicine, Utrecht University, Utrecht, The Netherlands, 7 Department for Infectious Diseases, Molecular Virology, Heidelberg University, Heidelberg, Germany, 8 Division Virus-Associated Carcinogenesis, German Cancer Research Center, Heidelberg, Germany, 9 University Medicine Greifswald, Institute of Bioinformatics, Greifswald, Germany

* niko.beerenwinkel@bsse.ethz.ch

**Data Availability Statement:** All relevant data are available and incorporated into the Supporting Information files or the software package

## Abstract

Genetic perturbation screens using RNA interference (RNAi) have been conducted successfully to identify host factors that are essential for the life cycle of bacteria or viruses. So far, most published studies identified host factors primarily for single pathogens. Furthermore, often only a small subset of genes, e.g., genes encoding kinases, have been targeted. Identification of host factors on a pan-pathogen level, i.e., genes that are crucial for the replication of a diverse group of pathogens has received relatively little attention, despite the fact that such common host factors would be highly relevant, for instance, for devising broad-spectrum anti-pathogenic drugs. Here, we present a novel two-stage procedure for the identification of host factors involved in the replication of different viruses using a combination of random effects models and Markov random walks on a functional interaction network. We first infer candidate genes by jointly analyzing multiple perturbations screens while at the same time adjusting for high variance inherent in these screens. Subsequently the inferred estimates are spread across a network of functional interactions thereby allowing for the analysis of missing genes in the biological studies, smoothing the effect sizes of previously found host factors, and considering a priori pathway information defined over edges of the network. We applied the procedure to RNAi screening data of four different positive-sense single-stranded RNA viruses, Hepatitis C virus, Chikungunya virus, Dengue virus and Severe acute respiratory syndrome coronavirus, and detected novel host factors, including UBC, PLCG1, and DYRK1B, which are predicted to significantly impact the replication cycles of these viruses. We validated the detected host factors experimentally using pharmacological inhibition and an additional siRNA screen and found that some of the predicted host factors indeed influence the replication of these pathogens.

accompanying the manuscript (https://github.com/cbg-ethz/perturbatr).

**Funding:** This work has been funded by ERASysAPP, the ERA-Net for Applied Systems Biology, under grant ERASysAPP-30 (SysVirDrug). ERASysAPP had no role in data collection, analysis or preparation of the manuscript.

**Competing interests:** Enter: The authors have declared that no competing interests exist.

## Author summary

Owing to their small genomes, positive-sense single-stranded RNA (ssRNA) viruses rely heavily on host factors, i.e. genes of the host species that either promote or inhibit viral replication. The identification of host factors that are essential for viral replication is not only of scientific interest, but also of clinical relevance, since they could serve as targets for the development of antiviral therapies, which is still unavailable for many important pathogens. So far genetic perturbation screens, for instance using RNAi or CRISPR, have been used to detect genes that influence the viral replication cycle. In these screens, host genes are first deprived of their function via genetic perturbation, followed by viral infection and quantification of viral replication. Finally the impact of identified genes on the replication cycle of said virus is assessed statistically. In the case of positive-sense ssRNA viruses a variety of such host factors have been earlier described and experimentally verified. However, most of the experiments have only analyzed a single virus. Since the majority of positive-sense ssRNA viruses have remarkably similar genomes and life cycles, we hypothesized that it should be possible to infer genes that restrict or promote replication of these viruses alike, allowing for the design of broad-spectrum drugs that target the entire group of viruses. Here, we present a two-stage procedure for broadly acting host dependency and restriction factor prioritization.

## Introduction

Genetic perturbation screens, such as RNA interference (RNAi) and CRIPSR-Cas9 screens, allow for the detection of host dependency and restriction factors by perturbing a target gene or transcript and observing its impact on the life cycle of a pathogen. In RNAi screens, genes are perturbed with small interferring RNAs (siRNAs). These are 20-25 nucleotides in length, complementary to mRNAs, and cause post-transcriptional gene silencing [1, 2]. The absence of certain host proteins has been shown to have an impact on the life cycle of pathogens [3, 4, 5], e.g., by reducing the ability of the pathogen to grow or by enhancing it.

Positive-sense ssRNA viruses (in the following also called group IV viruses according to the Baltimore classification [6]) such as the Hepatitis C virus, all share some common steps in their replication cycle. First, the virus enters the host cell and releases its RNA genome into the cytoplasm. Translation of the RNA results in the expression of viral (nonstructural) proteins that assemble into a replication complex that drives the synthesis of new viral RNA. Newly synthesized genomic RNA is encapsulated by capsid protein. Eventually, new virions are assembled and released from the infected cells [7, 8, 9]. For virtually all of these steps, the virus strongly depends on host proteins due to the small RNA virus genomes with limited coding capacity. Another common feature of +RNA viruses is that their RNA synthesis takes place in specialized structures that are associated with modified host membranes [10]. In order to understand the virus-host interplay reliable identification of potential host factors involved in virus replication is crucial.

However, statistical inference of these host factors is for multiple reasons often complicated. For example, siRNA-mediated knockdown can cause off-target effects such that often not only the transcript of interest is degraded but also other transcripts resulting in a non gene-specific phenotype [11, 12, 13, 14]. Furthermore, in cell-based assays different cellular states or cell context might lead to heterogeneous readouts [15, 16, 17].

So far statistical identification of host factors has either been conducted for single viruses [4, 8, 18, 19, 20], for two viruses of the same genus [5, 21] or family [22, 23], or for a group of

only very remotely related pathogens [24]. Prioritizing host factors on a viral group level, such as the group of positive-sense ssRNA viruses, has until now not been pursued in detail, even tough it seems promising, because viruses of the same group often have very similar replication cycles. Pathogens of one group might utilize the same, or at least functionally related, host factors and cellular pathways for replication. Consequently, development of anti-viral drugs targeting common host factors would have the potential for broad-spectrum activity. Despite its potential there are only very few pan-viral drugs under clinical investigation, for instance inhibitor development for PI4K$\beta$ targeting various human enteroviruses [25]. One of the reasons could be that the overall success rate for inferring pan-viral hits seems to be low, since even for single viruses the identified host or restriction factors have shown to be highly variable between different studies (e.g. between [22] and [23]). Interestingly, if hits found against one virus are tested against other viruses of the same group, it may well be observed that they are effective in the other viruses as well [23], which speaks for the hypothesis that analyses on a pathway-level could be promising or even necessary approaches.

Yet in most studies, statistical analysis is limited to gene- or siRNA-wise hypothesis tests, e.g., using t-tests or hyper-geometric tests [26, 27, 28, 29], not considering a priori information, for example, using biological networks, such as protein-protein interaction networks or co-expression networks. Network approaches have admittedly been used for various gene prioritization tasks [30, 31, 32, 33], but so far have found only little attention in virology. For instance, Maulik *et al.* [34] have presented a clustering approach to detect modules in a bipartite viral-host protein-protein interaction network to identify host factors. Amberkas *et al.* use a meta-analysis approach using network modules for RNAi screens [35]. Wang *et al.* [36] use a scoring system based on integration of several RNAi screens to account for false positives and negatives. However, while these approaches include a priori knowledge, they cannot be used to detect genes on a pan-pathogen level.

Here, we present a two-stage procedure for pan-pathogen host dependency and restriction factor identification (Fig 1), and apply it to RNAi screening data sets comprising four different positive-sense ssRNA viruses, i.e. Hepatitis C virus (HCV), Chikungunya virus (CHIKV), Dengue virus (DENV) and SARS-coronavirus (SARS-CoV). First, we apply a maximum likelihood approach for joint analysis of viral host factors using a random effects model. Then, we propagate this information over a biological graph using network diffusion with Markov random walks in order to account for genes of importance on a pathway level, reduce the number of false negatives and possibly stabilize the ranking of host factors. With our approach it is possible to detect novel pan-pathogen host factors, while also considering prior information in the form of networks. Our model has been designed for heterogeneous data sets by accounting for various confounding factors within the data. When applying our method to six different RNAi screening data sets of the four positive sense ssRNA viruses, CHIKV, DENV, HCV and SARS-CoV, we found that the procedure is able to recover the host factors for single viruses that have been described in the literature before, and to predict novel pan-pathogen host factors. We validated the host factors for which compounds were commercially available experimentally using pharmacological inhibition screens for five virus, i.e., HCV, DENV, CHIKV, Middle-East respiratory syndrome coronavirus (MERS-CoV) and Coxsackie B virus (CVB). Moreover, we validated the newly predicted host factors, UBC, EP300 and PLCG1, using another siRNA knockdown on the Hepatitis C virus.

## Methods

In this section, we introduce the two-stage procedure which is then applied for inferring pan-pathogen host factors. The first part of the procedure consists of a random effects model that is

**A** **Integrate and normalize RNAi data sets from four positive-sense ssRNA viruses.**

**CHIKV**

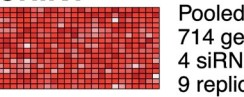

Pooled screen
714 genes
4 siRNAs per gene
9 replicates per siRNA

**DENV**

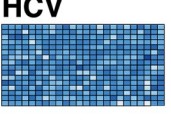

Arrayed screen
714 genes
2–6 siRNAs per gene
5 replicates per siRNA

**HCV**

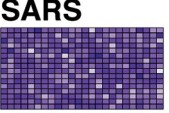

Arrayed screen
714 genes
2–6 siRNAs per gene
12 replicates per siRNA

**SARS**

Pooled screen
714 genes
4 siRNAs per gene
9 replicates per siRNA

**B** **Model the readout of the pan-viral data set as a linear combination of fixed and random gene effects.**

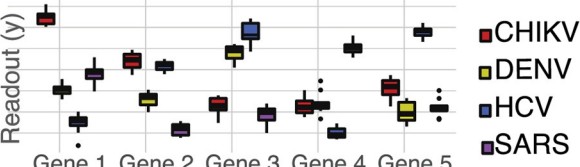

■CHIKV
■DENV
■HCV
■SARS

**Estimation of random gene effects $\gamma$**

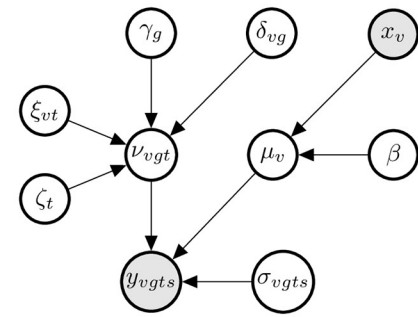

**C** **Include prior knowledge: diffuse estimated gene effects over a PPI network.**

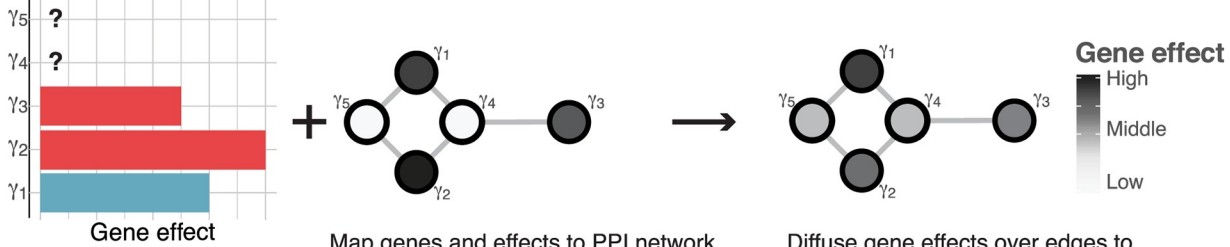

Map genes and effects to PPI network and normalize gene effects.

Diffuse gene effects over edges to re-rank and consider missing genes.

Gene effect
High
Middle
Low

**Fig 1. Integrated host factor prioritization from viral infection RNAi screening data using a two-stage procedure.** (A) We normalized and integrated data from RNAi perturbation screens of four different positive-sense RNA viruses. (B) Stage 1: We estimate pan-viral effects $\gamma = \{\gamma_1, \ldots, \gamma_G\}$ from the integrated data sets for each of $G$ genes using a random effects model and rank the genes by their absolute effect size. The gene effects represent the impact of a genetic knockdown of the life cycle on the entire group of viruses. (C) Stage 2: To account for genes that have not been knocked down in the RNAi screens, and to possibly account for false negatives in our rankings using biological prior knowledge, we map the gene effects $\gamma_g$ onto a protein-protein interaction network. We then propagate the inferred estimates over the graph using network diffusion resulting in a final ranking of genes that are predicted to have a significant impact on the pan-viral replication cycle.

used to infer pan-pathogen gene effects that quantify the overall impact of gene perturbation on the life cycle of a group of pathogens. The second part of the procedure uses the inferred gene effects and propagates them over a biological network.

## Random effects model

We model the readout $y_{vgts}$ of an integrated perturbation screen for virus $v$, gene $g$, siRNA $s$ and stage of infection $t$ using a linear random effects model, where different intercept terms for different biological hierarchies (groups) are introduced. Stage $t$ is introduced to distinguish effects that are primarily due to early stages of the viral replication cycle (entry and replication) vs later stages (assembly and release).

RNAi perturbation screens often suffer from high variability between replicates [11, 29, 37, 38]. To account for this variability, we introduce four random intercept terms that correct for differences in the variance of genes, viruses, and infection stages. The remaining variance that is not explained by these random intercepts or the fixed effects is captured by a Gaussian error term. The readout is modeled as

$$y_{vgts} = x_v \beta + \gamma_g + \delta_{vg} + \zeta_t + \xi_{vt} + \epsilon_{vgts}, \tag{1}$$

where the random effects and noise are distributed as

$$
\begin{aligned}
\gamma_g &\sim \mathcal{N}(0, \sigma_\gamma^2), \\
\delta_{vg} &\sim \mathcal{N}(0, \sigma_\delta^2), \\
\zeta_t &\sim \mathcal{N}(0, \sigma_\zeta^2), \\
\xi_{vt} &\sim \mathcal{N}(0, \sigma_\xi^2), \\
\epsilon_{vgts} &\sim \mathcal{N}(0, \sigma_\epsilon^2).
\end{aligned}
$$

The covariate $x_v$ is a categorial variable representing the virus type using treatment contrasts, $\beta$ is a fixed effect coefficient, $\gamma_g$ and $\delta_{vg}$ are random effects for genes and nested effects for genes within each virus, respectively. The terms $\zeta_t$ and $\xi_{vt}$ describe random effects for infection stages and nested effects for infection stages within each virus, respectively. The remaining noise of the model is captured by $\epsilon_{vgts}$.

The random effects model is fitted using the R-package lme4 [39] using weighted restricted maximum likelihood (Supplement S6 Text for details).

**Gene effect ranking.** The model defined in Eq (1) allows identification of potential host dependency and restriction factors on a pan-pathogen level, i.e., detection of host genes that potentially alter and impact pathogen growth. The strength of the effect of a gene knockdown (the effect size) on the replication cycle of a group of pathogens is given by the estimated random effect $\gamma_g$ for a gene $g$. A negative gene effect $\gamma_g < 0$ means that knockdown of gene $g$ restricts viral replication. A positive gene effect $\gamma_g > 0$ means that knockdown of gene $g$ promotes viral replication. Furthermore, we estimate the pathogen-specific gene effect as $\rho_{vg} = \gamma_g + \delta_{vg}$ [24].

## Gene effect network propagation

We employ network diffusion to inform our estimates on a pathway-level post-inference and in order to account for host genes missing in the analysis (for instance, unscreened genes), potential false negatives, and to stabilize gene rankings using prior information. The diffusion is used after estimation of gene effect sizes using the random effects model from Eq (1). The Markov random walk is applied over a network of genes where edges represent biological relationships. These relationships can, for example, be encoded as interaction strengths between proteins, gene co-expression patterns, or common transcription factor binding sites. Using network diffusion it is possible to spread the information of single starting nodes, i.e. genes for which gene effects $\gamma_g$ have been estimated (Eq (1)), to their surrounding neighbours to include potential genes in the list of host factors, reduce the number of false negatives and stabilize the predicted ranking of genes given by their effect strengths $\gamma_g$.

Instead of choosing neighbors of a gene directly which would potentially introduce false positives, Cowen *et al.* [31] argue that a diffusion approach has the advantage of down-weighing new predictions that are only supported by few edges or edges with low weight. Furthermore, genes that are connected to the prior list of genes by several edges or edges with high weights have stronger support.

We initialize the starting distribution over $N$ network nodes of the Markov chain as:

$$\mathbf{p}_0 = \left( \frac{|\gamma_1|}{\sum |\gamma_i|}, \ldots, \frac{|\gamma_G|}{\sum |\gamma_i|}, 0, \ldots, 0 \right)^T, \qquad (2)$$

where $G \leq N$ is the number of genes estimated using Eq (1), i.e. the number of genes with estimated effects $\gamma_g$. Using $\mathbf{p}_0$ the Markov chain is run until convergence with updates,

$$\mathbf{p}_t = (1 - r)\mathbf{W}\mathbf{p}_{t-1} + r\mathbf{p}_0, \qquad (3)$$

where $r$ is a user-defined restart probability, i.e., the chance that the random walk returns to its initial state and $\mathbf{W}$ is a left stochastic transition matrix derived from a biological network. In this study we use the functional protein interaction network from [40]. They define a functional interaction as *one in which two proteins are involved in the same biochemical reaction as an input, catalyst, activator, or inhibitor, or as two members of the same protein complex*, i.e. functionally significant molecular events in cellular pathways and not mere protein-protein interactions which rarely show direct evidence of being involved in biochemical events. The network consists in part of expert-curated, high-quality functional edges and in part of edges that have been trained and validated with a naive Bayes classifier. Unlike many other biological networks, the high quality of the annotations does not necessitate choosing edges with care, such as edges derived from computational annotation or inference with older yeast-two-hybrid technologies which are frequently false positives. Moreover, due to the biological interpretability of the edges in a pathway-context, a functional network like this should serve as a good choice to infer novel restriction and dependency factors and stabilize our rankings, because it associates genes connected with a disease and separates genes with mere physical interaction as in conventional pairwise networks. We stochastically normalized the weighted adjacency matrix of this network and then use the normalized matrix as transition matrix $\mathbf{W}$. After convergence of the Markov chain, we use its stationary distribution $\mathbf{p}_\infty$ as new ranking of host factors by sorting genes accordingly.

For a random walk on a network that uses restarts, the length of the walk, $l$, i.e., the number of edges it travels, can be modelled as a geometric random variable:

$$P_r(l) = (1 - r)^{l-1} r, \ l \in \{1, 2, \ldots\}$$

that is parametrized by a success probability $r \in [0, 1)$, and models the number of Bernoulli trials $l$ needed for a success. The mean of the geometric distribution $\mathbb{E}[l] = \frac{1}{r}$ directly relates to the average length of the random walk. For instance, choosing a success probability of $r = 0.5$ would result in on average 2 trials until success. For a success probability of $r = 0.2$ the average number of trials is $\mathbb{E}[l] = 5$, which yields an average path length of 5. Consequently, choosing a high success probability reduces the average number of edges travelled automatically and ranks the starting genes higher than genes farther away. We chose to use a restart probability of $r = 35\%$, opting for on average approximately 3 travelled edges. Restart probabilities higher than 50% deprioritize the network information over the data, while lower restart probabilities than 20% give too much weight to the prior knowledge.

## Model assessment

We validate our method on simulated as well as biological data. First we conduct analyses of the stability of gene rankings that our two-stage procedure produces. Then we assess the predictive performance of the random effects model in comparison to another model (PMM [24]). Here, we briefly describe the procedures to simulate data and the used methods for assessment.

**Data simulation.** We simulated data using the procedures described in Supplement S2 and S3 Texts. Briefly, we sampled random vectors of effects for genes, viruses and screen types and took all possible combinations over the three random vectors. Then, we replicated every observation 8 times to guarantee convergence of the solver and added normal i.i.d. noise to every observation. We created three data sets and added low, medium, and high i.i.d. white noise ($\epsilon \sim \mathcal{N}(0, \sigma^2)$, $\sigma^2 \in \{1, 2, 5\}$), respectively, separately to every observation.

**Performance measures for stability analysis.** We boostrap every simulated data set or biological 10 times. For every bootstrap sample we sort the gene effects from the hierarchical model by their absolute effect sizes and the equilibrium distributions of the network diffusion. For every bootstrap sample $j$ we take the top $n \in \{10, 25, 50, 75, 100\}$ gene effects as well as the top $n$ equilibrium probabilities. We then take each pair $(j, k)$ of bootstrap samples and compare the top $n$ gene effect vectors and highest $n$ equilibrium probability vectors. For every pair $(\mathcal{A}, \mathcal{B})$ of the top $n$ elements of either gene effects or equilibrium distributions, we compute the Jaccard index as $J(\mathcal{A}, \mathcal{B}) = \frac{|\mathcal{A} \cap \mathcal{B}|}{|\mathcal{A} \cup \mathcal{B}|}$ and Spearman's correlation coefficient (Supplement S2 Text and Supplement S1 Code).

**Performance measures to assess predictive performance.** We use 10-fold cross-validation in order to assess the predictive performance between our random effects model (Eq (1)) and PMM. We repeatedly split the data in training and test sets and iteratively trained on nine folds and predicted gene effects on the test fold. Finally, we compute the mean squared error for every fold for each of the two models (Supplement S3 Text and Supplement S1 Code).

## Results

We applied our method for gene prioritization to six biological data sets of four positive-sense ssRNA viruses, HCV, DENV, CHIKV and SARS-CoV, and inferred potential pan-viral dependency and restriction factors. We then validated the highest ranked host factors. We first show results for normalization and integration of the RNAi data sets, then present the application of the procedure and a benchmark, and finally discuss the biological findings. The entire procedure is implemented in an R-package called `perturbatr` available on Bioconductor.

### Data sets and normalization

We integrated data from six RNAi perturbation screens consisting of the four positive-sense ssRNA viruses HCV, DENV, CHIKV and SARS-CoV. These screens have been generated under different biological conditions (Table 1). Following the definition in Eq (1), we distinguish different stages of infection, i.e., either 'early' when the screen was conducted for detection of host factors that are essential for viral entry and replication, or 'late' when the host factors are required for viral assembly and release. Screening of ssRNA viruses has been conducted on MRC5 cells for CHIKV, Huh7 cells for DENV, Huh7.5 cells for HCV, and 293/ACE2 cells for SARS-CoV. The screens used either libraries of Dharmacon SMART-pools (4 siRNAs per well/gene) for CHIKV and SARS-CoV or unpooled Ambion libraries for HCV and DENV. We filtered the six RNAi data sets for genes that are available for every virus which left a data set with a total of 714 genes and controls (Fig 1). For each of the screens, siRNAs have been placed on 384-, or 96-well plates, respectively. Cells have been seeded and, after transfection with siRNAs, infected with the respective reporter virus (Table 1). Univariate readouts are either measurements of viral or reporter protein (GFP/Luciferase).

In order to have comparable phenotypes, i.e., fluorescence and luciferase readouts, special emphasis has to be put on normalizing the screens, because different cell types (MRC5/Huh7/Huh7.5/293ACE2) can lead to slightly different gene expression and knockdown patterns. Furthermore, in addition to high between-screen variability in RNAi perturbations, high variance

**Table 1. Meta data of positive-sense ssRNA viral RNAi screens.** The data sets are derived from separate screens using different cell lines, readout types or infection stages. We use six RNAi screens for Chikungunya virus, Dengue virus, Hepatitis C virus, and SARS coronavirus.

| Virus | Stage | Cell type | Readout | Library | Screen | Reference |
|---|---|---|---|---|---|---|
| CHIKV | early | MRC5 | GFP | Dharmacon pool | Kinome | |
| DENV | early | Huh7 | E-protein | Ambion single | Kinome | Cortese et al. [41] |
| DENV | early | Huh7 | Luciferase | Ambion single | Genome | |
| DENV | late | Huh7 | Luciferase | Ambion single | Genome | |
| HCV | early | Huh7.5 | GFP | Ambion single | Kinome | Reiss et al. [8] |
| HCV | early | Huh7.5 | Luciferase | Ambion single | Genome | Poenisch et al. [19] |
| HCV | late | Huh7.5 | Luciferase | Ambion single | Genome | Poenisch et al. [19] |
| SARS-CoV | early | 293/ACE2 | GFP | Dharmacon pool | Kinome | De Wilde et al. [4] |

between plates from the same screen has to be taken into consideration (Fig 2). Before normalization plates are not comparable due to highly varying plate effects (Fig 2a). After normalization the data are in a final step centered and scaled to unit variance yielding comparable phenotypes (Fig 2b).

High variability of phenotypes is mainly due to batch effects, stochasticity in transfection and knockdown, and spatial effects in rows and columns, i.e., when wells on the margin on average have higher or lower readouts compared to wells in the center. To account for these effects, we use a combination of different normalization techniques for every screen separately (Supplement S1 Text for details). Briefly, the CHIVK and SARS-CoV screens use a pooled Dharmacon library on 96 well plates. We normalized the two data sets by first taking the natural logarithm over all samples, then substracting the mean background signal and finally computing a robust Z-score over the whole plate readout. The procedure has been applied separately for every plate. Since genes were not randomized on plates we did not use B-scoring or other methods that account for spatial effects [2, 29, 38]. For the HCV and DENV genome screens, we computed the natural logarithm for every readout of the complete data set, B-scored the plates using two-way median polish and, in a last step, calculated robust Z-scores. The HCV and DENV kinome screens have been normalized by first taking the natural logarithm of the well readouts and then fitting a local regression model to correct for cell counts. Since the HCV and DENV screens have randomized plate designs, we also corrected for spatial

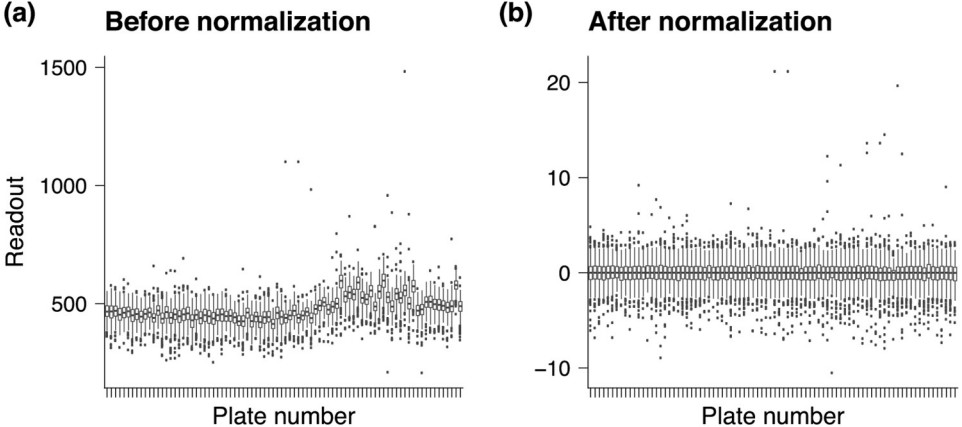

**Fig 2. Comparison of readouts for unnormalized vs. normalized data.** Every box-plot shows the distribution of readouts of a single plate on the x-axis. (a) Before normalization between plate readouts are hardly comparable due to batch and spatial effects. (b) After normalization the data are eventually centered and scaled to unit variance yielding comparable phenotypes.

effects using two-way median polish using B-scores and eventually computed robust Z-scores (see Supplement S1 Code for the exact procedures).

## Model assessment

In order to assess the stability of the inferred gene effects $\gamma_g$ and the equilibrium distribution $p_{\infty_g}$, and the predictive power of the random effects model, we applied the model to two different data sets: a simulated data set and the biological data set described before. First, we compare the stability of hits, i.e., if the same genes appear among bootstrapped data sets, using the Jaccard index and Spearman's correlation coefficient. Second, we assess the predictive power of the model using cross-validation.

**Stability analysis.** The models described by Eqs (1) and (3) estimate gene effects $\gamma_g$ and an equilibrium distribution $p_{\infty_g}$ for every gene $g$. To assess the reproducibility of these estimates, i.e., the consistency of the rankings of gene effects and equilibrium distributions, we applied the model to several simulated data sets as well as to the pan-viral biological data set introduced above.

*Simulated data.* We simulated data as described before and validated the consistency of the rankings of these data sets (Fig 3a). For low error variances the stability of both the random effects model and the network diffusion is high between bootstrap samples. Increasing the error levels for the hierarchical model only seems to reduce the Jaccard index, while the Spearman correlations are staying stable. For high error levels and the first $n = 10$ genes, two sets of bootstrap samples have on average 60% similarity and a correlation of around 90% for the random effects model. The network diffusion, on the other hand, seems to be robust to increasing error variances having similar Jaccard indexes and correlation for medium and high error variance, emphasizing the previous argument regarding the stabilizing function of the network diffusion.

*Biological data.* We performed a similar analysis on the biological data set. Instead of comparing different noise levels we validated how the number of examined viruses influences the different rankings. We bootstrapped the data set again and computed the Jaccard index and Spearman's correlation coefficient for every pair of bootstrap samples. For both models, increasing the number of viruses from 2 to 4, does not significantly alter the Jaccard indexes for all numbers of genes (Fig 3b). However, increasing the number of viruses reduces correlations for both models. While the reductions are only marginal for higher gene numbers for the random effects model, they are stronger for the network diffusion. Lower correlations can be explained by the fact that RNAi screens are highly variable and different bootstrap samples give as a consequence varying estimates of gene effects.

**Analysis of predictive performance.** In order to validate the predictive performance of the random effects model from novel data, we used a simulated data set and the biological data set as before, and benchmark the predictive performance using 10-fold cross-validation. We compare our method against another random effects model, called PMM [24].

*Simulated data* We created three data sets using the procedure described in Supplement S3 Text. As before, the data sets can be distinguished by the amount of noise that has been added to every observation. Our hierarchical model consistently outperforms PMM for different levels of variance and different validation methods (Supplement S4a Fig). This is largely due to the fact that our model was tailored to considering heterogeneous RNAi screens where different infection stages are present while PMM does not make this distinction.

*Biological data* For the biological analysis we used the integrated pan-viral RNAi screen as before. In this benchmark, our model slightly outperforms PMM (Supplement S4b Fig). Our model achieves a lower mean residual sum of squares on all test sets. Furthermore, increasing the number of viruses from two to four, leads to a decrease of mean residual sum of squares.

**(a)** Synthetic data benchmark

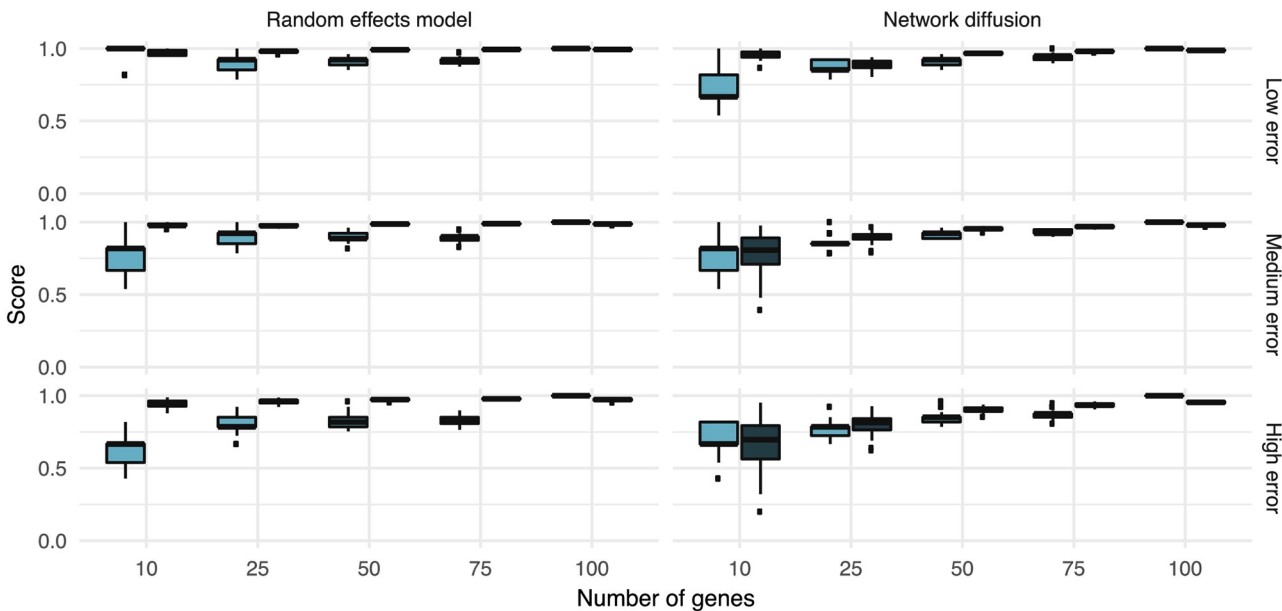

**(b)** Biological data benchmark

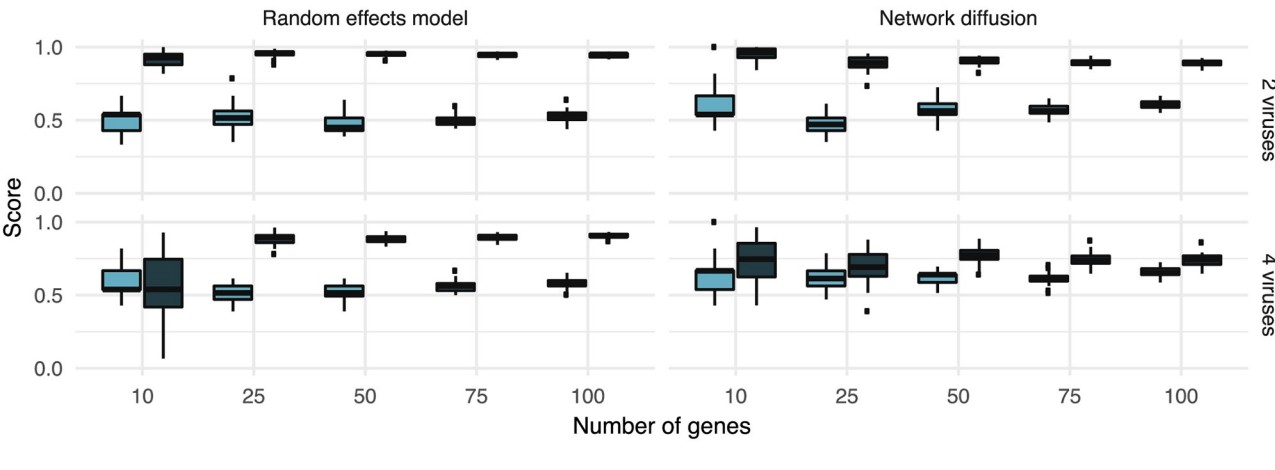

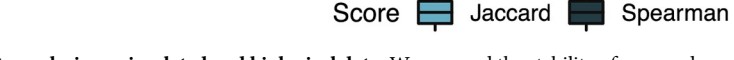

**Fig 3. Stability analysis on simulated and biological data.** We assessed the stability of our random effects model using the Jaccard index and Spearman's correlation coefficient (y-axis) given the first $i \in \{10, 25, 50, 75, 100\}$ highest ranked genes from 100 bootstrap samples (x-axis). (a) For low error variance $\sigma^2 = 1$, gene rankings are highly stable. While increasing the error variance keeps correlations stable, Jaccard indexes reduce. The network diffusion is stable against increasing error variances having similar Jaccard indexes and correlation for medium and high error variance. (b) On the biological data set increasing the number of viruses does not significantly reduce Jaccard indexes or correlations for the random effects, with the exception for the correlations for 10 genes. The network diffusion has stable Jaccard indexes for increasing virus numbers at around 60%. The correlations between bootstrap samples, however, decrease with a higher number of viruses.

## Biological results

We applied our method to the RNAi screening data described in Table 1. First we estimated gene-effects $\gamma_g$ using the random effects model described in Eq (1) and then propagated these

**Table 2. First 20 host dependency and restriction factors selected by the ranking of the network diffusion using a restart probability of $r = 0.35$.** 'Ranking' shows the rank after network diffusion, 'Gene effect' shows the effect sizes $\gamma_g$ inferred by the hierarchical model, the other columns show virus specific effects $\rho_{vg}$.

| Gene | Ranking | Gene effect | CHIKV | DENV | HCV | SARS-CoV |
|---|---|---|---|---|---|---|
| ubc | 1 | n.a. | n.a. | n.a. | n.a. | n.a. |
| plk1 | 2 | -0.14 | -0.11 | -0.25 | -0.15 | -3.77 |
| dyrk1b | 3 | -0.15 | -0.26 | -0.14 | -0.15 | -3.97 |
| pik4ca | 4 | -0.13 | -0.12 | -0.48 | -2.69 | -0.51 |
| mapk3 | 5 | -0.06 | -0.52 | -0.33 | 0.05 | -0.93 |
| pik3r1 | 6 | 0.06 | 0.47 | 0.36 | 0.39 | 0.51 |
| dusp1 | 7 | -0.12 | -1.45 | 0.02 | -0.13 | -2.05 |
| pck1 | 8 | -0.11 | -0.69 | -0.84 | -0.80 | -1.08 |
| ep300 | 9 | n.a. | n.a. | n.a. | n.a. | n.a. |
| mapk1 | 10 | 0.03 | -0.09 | -0.14 | 0.30 | 0.74 |
| pkn3 | 11 | -0.11 | -0.11 | -0.63 | -0.22 | -2.37 |
| dgke | 12 | -0.11 | -1.16 | 0.02 | -0.13 | -1.98 |
| plcg1 | 13 | n.a. | n.a. | n.a. | n.a. | n.a. |
| cdk6 | 14 | 0.08 | 0.16 | 0.70 | 0.47 | 1.22 |
| lats1 | 15 | 0.08 | 0.47 | 0.41 | 0.46 | 0.99 |
| csnk2b | 16 | -0.10 | -1.02 | -0.82 | -0.46 | -0.62 |
| cdk5r2 | 17 | -0.10 | 0.78 | -1.51 | -0.10 | -2.19 |
| shc1 | 18 | -0.07 | -0.97 | -1.24 | -0.63 | 0.76 |
| mapk14 | 19 | 0.03 | -0.06 | 0.15 | 0.25 | 0.59 |
| camkk2 | 20 | -0.10 | -1.66 | -0.85 | -0.48 | 0.09 |

effects over a functional protein-protein interaction network using network diffusion (Eq (3)) resulting in a ranking of genes by their estimated impact on the life-cycle of the group of viruses. We validated the inferred genes for five viruses using pharmacological inhibition and another siRNA knockdown of three further genes for HCV.

**Gene effect ranking.** Given the results from the stability analysis and analysis of predictive performance, we concluded that the proposed random effects model model is preferable to PMM, due to the fact that it captures more of the variance in the data, for instance, when strong infection stage effects are visible, and because it allows distinguishing between genes that are influencing the viral replication cycle in the early stages of replication, or in the later stages, respectively.

We applied the hierarchical model to the pan-viral data set and inferred the gene effects $\gamma_g$ (of which the top 25 are shown in Supplement S5 Text). We then used the estimated gene effects $\gamma_g$ and propagated these using the Markov random walk described in Eq (3). After diffusion we obtain a ranking of all genes in the network (Table 2). While the majority of genes has already been previously selected by the random effects model, we also discovered novel hits, such as UBC (rank 1), EP300 (rank 9), and PLCG1 (rank 13) using the network diffusion. Among the strongest effectors derived from the hierarchical model are, DYRK1B (rank 3), a nuclear-localized protein kinase participating in cell-cycle regulation, and PKN3 (rank 11), a rather little studied kinase that has been implicated in Rho GTPase regulation and PI3K-Akt signaling. UBC encodes ubiquitin, which is involved in numerous cellular processes, most prominently protein degradation. PLCG1 is crucially involved in signal transduction from receptor-mediated tyrosin kinases (e.g. Src) and catalyzes the formation of the second messenger IP3 and DAG. Recently PLCG1 was also found to impact progression of HCC [42], the HCV replication cycle [43], as well as receptor-mediated inflammation and innate immunity

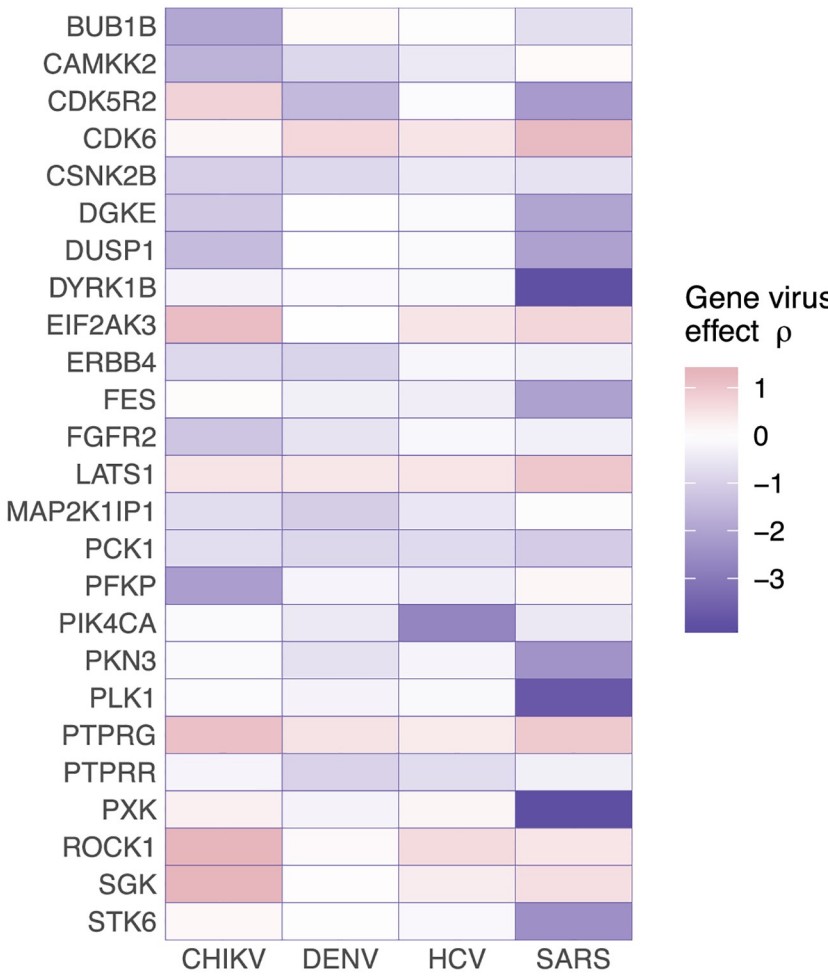

**Fig 4. Effect matrix of pathogen-specific gene effect strengths $\rho_{vg}$.** The 25 strongest hits when sorting by absolute effect sizes $\gamma_g$ are shown. Every column shows one virus and every row represents the effect size of a gene knockdown on the specific virus $\rho_{vg}$. For some of the genes, such as DYRK1B, PKN3, CDK6 or CSNK2B, the knockdown has an either all-positive, or all-negative effect on the viral replication cycle.

[44]. EP300 is an acetyltransferase and acts as a transcriptional co-activator and has not been studied in detail so far.

We compared the strongest gene effects $\gamma_g$ inferred by the hierarchical model (Supplement S5 Fig) to the virus-specific gene effects $\rho_{vg}$ (for which RNAi screens have mostly been used; Fig 4) and found that for some of the estimates for the gene effects $\gamma_g$ the pathogen-specific effects are not consistent over all pathogens. For example, while perturbation of gene CDK5R2 has a beneficial impact on CHIKV replication, it has a restricting effect on the other three viruses. On the other hand perturbation of DYRK1B, PKN3, CDK6, or CSNK2B has either an all-negative or all-positive impact on the replication cycle of the ensemble of viruses. Genes that upon perturbation show the same consistent effect, i.e. suppression of early or late stages of the viral replication cycle, could be targets for the development of broad-spectrum antiviral drugs.

**Validation of identified host factors.** We validated some of the top genes from Table 2 using pharmacological inhibitors to verify whether the predicted genes are indeed host factors

that are involved in viral replication. In short, we searched the literature for inhibitors and conducted a screen for the proteins for which compounds were commercially available (see Supplement S4 Text for details on the experimental setup and Supplement S6 Fig for results). In order to assess if the top inferred gene products really have a pan-viral effect, inhibitors were tested on DENV, CHIKV and HCV as before and two novel positive-strand ssRNA viruses, MERS-CoV and CVB. Of the top 20 host factors from Table 2 inhibitors were available for the dependency factors CAMKK2, CDK5R2, DGKE, DUSP1, DYRK1B, PIK4CA, PKN3 and PLK1. The inhibitors were tested in dose-response CPE reduction assays on cells infected with the viruses. In parallel we assessed cytotoxicity of the compounds and discarded measurements that led to a significant reduction in cell viability (below 75% of the signal obtained for untreated control cells). For every host factor, virus and compound concentration, we tested if inhibition of a protein reduced viral replication in comparison to a negative control significantly (one-sided two-sample Wilcoxon test). We adjusted all $p$-values for multiple testing using the Benjamini-Hochberg correction [45]. We found that inhibition of several host factors showed significant reductions in replication on subsets of the five viruses and specific compound concentrations. For instance, CDK5R2, PKN3 and DYRK1B were significant at the 10%-level after multiple testing correction for at least some compound concentrations in four of the five viruses. However, none of the tested compounds had a significant effect on the replication of all of the five viruses (Supplement S6 Fig). Note that PLK1 was discarded due to cytotoxicity of the inhibitor at higher compound concentrations. For that reason, we point out that PLK1 should possibly also be discarded in the analysis of the primary screens.

Furthermore, we validated the three genes that were newly identified by the network model (UBC, PLCG1, EP300) for HCV using two different siRNAs per gene. In particular, we were interested to see whether knockdown of these three genes would impact the viral replication significantly (see Supplement S5 Text for experimental details, data normalization and statistical analysis). We found that knockdown of UBC and PLCG1 caused a significant inhibition of replication at a level of $\alpha = 5\%$ (Fig 5) in comparison to a negative control for all tested siRNAs (two-sided two-sample Wilcoxon-test). However, EP300 was not confirmed at the same significance level for both siRNAs tested.

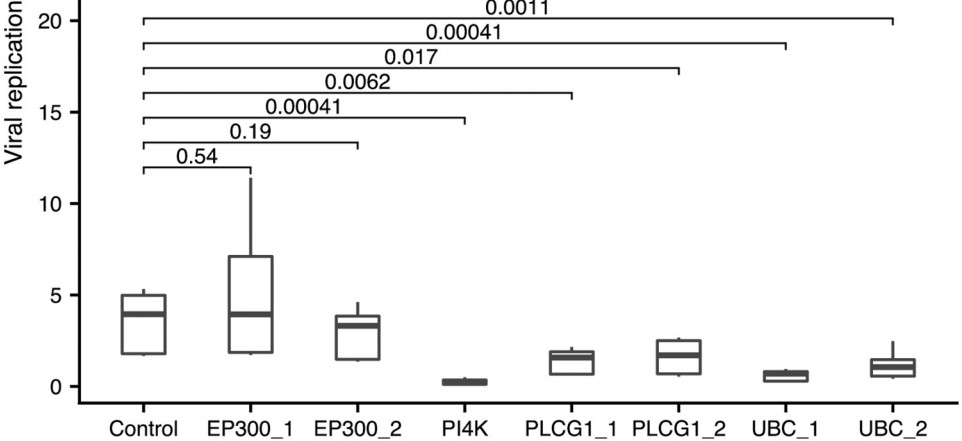

**Fig 5. Validation of UBC, PLCG and EP300 against a negative control and a positive control, PI4K, for HCV.** UBC and PLCG1 show significant $p$-values at the 5%-level for all validated siRNAs. The positive control PI4K also was highly significant, while the two siRNAs used for EP300 did not show a significant trend.

## Discussion

In this work, we have integrated RNAi screening data of a group of four different positive-sense ssRNA viruses and presented a two-stage procedure to prioritize pan-viral host dependency and restriction factors from genetic perturbation screens. The result of our method is a ranking of genes that are predicted to impact the life cycle of an entire group of pathogens. We implemented the two-stage procedure in an R-package called `perturbatr` which is designed for the analysis of large-scale high-throughput perturbation screens of multiple data sets and is available on GitHub and Bioconductor.

We validated host factors for which pharmacological inhibitors were commercially available experimentally by treating cells infected with five positive-sense ssRNA viruses with these compounds, and another siRNA knockdown of the three newly predicted genes on HCV.

Our procedure first infers a list of possible host factors using a random effects model where we model the readout of a genetic perturbation screen as a linear dependency on a virus, a pan-viral gene effect $\gamma_g$, and a sum of other random effects to capture the heterogeneity of the data. With a likelihood-based formulation jointly analyzing genetic perturbation screens of different viral RNAi screens is straightforward in comparison to a meta-analysis, since in the latter case every virus is analyzed independently and results have to be aggregated, thereby potentially discarding common host factors. Furthermore, the noise model and inclusion of random effect terms allow to account for high variance in the data sets.

The list of gene effects $\gamma_g$ is then propagated over a functional interaction network using a Markov random walk with restarts. Functional interactions networks, such as [40], allow incorporation of true biological association in a pathway-context to the analysis and stabilizing of the the rankings. By subsequently applying a network diffusion approach it is also possible to not only account for genes that have not been in the primary RNAi screens, but also to re-rank genes using pathway information allowing to potentially reduce the number of false negative predictions.

The analysis produced a set of host factors, such as DYRK1B, UBC, PLCG1 and PKN3, that likely impact the replication cycle of a broad range of positive-sense ssRNA viruses. Of the top 20 host factors (Table 2), we were able to find commercially available compounds for nine of them, which we then biologically validated. While the screen confirmed the importance of these genes on the pan-viral replication cycle of subsets of viruses, no host factor could be found that is significant for all viruses. In general, viruses usurp defined cellular pathways. Even closely related viruses may use different entry points to the pathway. One example are the Dengue and Zika viruses which both depend on the host factor STT3A, but only DENV requires STT3B for replication [22]. The degree of similarity of the molecular biology of the viruses seems to determine the success of finding pan-viral genes in contrast to finding relevant pathways. While it makes theoretical sense that all positive-sense ssRNA viruses use the same host factors, detection of these has proven to be complicated and, as already mentioned in the introduction, yields variable results even for the same virus. A lack of overlap between screens, flexibility of the cell in several aspects and the possibility of viruses to just take different routes to achieve replication corroborates this hypothesis and makes pathway-analyses even more important. The broader the targeted group of viruses, the more central a target gene would have to be (e.g. UBC), but in that case it gets increasingly unlikely to find a inhibitor condition that only harms the virus but not the host cell. For bacteria, antibiotics are only specific to a more or less related group of bacteria (e.g. gram-positives), because of the metabolic similarity of the group. For viruses, it is likely that these groups need

to be much narrower because in many cases only closely related viruses might actually share enough similarity in the metabolic or regulators pathways they exploit. Additionally, it has to be emphasized though that a protein inhibition screen like the one we conducted is not perfectly able to validate the inferred genes and their function in the replication cycle of the viruses. Thus a more rigorous validation could shed light on the biological importance of these genes.

We validated the three newly found host factors, UBC, PLCG1, and EP300, using siRNA knockdown for HCV and could confirm UBC and PLCG1 to be proviral host factors. Generally, host dependencies and restriction factors are not necessarily crucial for host cells survival, i.e. host factors can be knocked down without inducing cell death. Exceptions are single candidates such as UBC which is central player in cell biology. Ubiquitination of proteins can target them for degradation in the proteasome which is an important homeostatic process in every cell. The proteasome has come up frequently as host factor for many viruses, albeit not always the same genes [46]. Inhibition of the proteasome, while being vital for the cell, is already done therapeutically, for instance in cancer treatment [47], or in studies for antiviral treatment [48]. Consequently, the inhibition of host factors that are also crucial for the host cell can be achieved even though it is a matter of fine balancing between cytotoxicity to the cell and efficacy against disease.

The proposed procedure to infer pan-pathogen host factors could aid in the development of broad-spectrum antiviral drugs for a group of viruses or even bacteria that could allow the treatment of multiple diseases (Table 2) with the same substance. In addition, our model generates estimates of gene effect sizes for the single viruses.

In this work we selected a group of positive-sense ssRNA for analysis. The replication cycles of any subgroup of positive-sense ssRNA viruses consist of notably similar steps and, given the similarities of how they replicate, we hypothesized that they share the same host dependency or restriction factors or, at least, the same pathways (hence the network analysis). While our model can be applied to any group of pathogens the success of finding relevant host-factors for a highly diverse group of pathogens is less unlikely. In addition the experimental design of such a study, a factor which we did not emphasize enough, is critical: contributing factors might be quality of interventions, number of replicates, or the type of readout, e.g. GFP signals of viral growth or cell death, or even sequencing data in CRISPR screens.

Our two-stage procedure has also some limitations. In our case the integrated data set showed strong heterogeneity and variance between the different biological conditions which necessitated the inclusion of random effects. For data sets with less variance a random effects model might not be needed at all. Moreover, utilizing biological prior knowledge in the form of protein-protein interaction networks could possibly bias and corrupt results, especially when networks with incorrect edges are used. The use of multiple, different networks may improve this situation [33].

Since we apply a stochastic approach for network diffusion we cannot gain information about whether genes are dependency or restriction factors. This could be addressed by developing a network diffusion model applying state probabilities for pro-viral and anti-viral effects. Finally our method does not provide estimates for statistical significance for the genes, but only a ranking of genes.

Currently our model can be used for RNAi screens with continuous readouts, but can readily be generalized to sequencing-based perturbation screening methods, such as CRISPR, where read counts are usually modelled as negative binomial or Poisson random variables.

## Supporting information

**S1 Code. Source code all relevant source files used for integration, normalization and analysis of data.**
(ZIP)

**S1 Text. Normalisation methods.** Description of methods we used for normalizing and integrating the data sets.
(PDF)

**S2 Text. Stability analysis.** Description of creation of the synthetic data sets for stability analysis.
(PDF)

**S3 Text. Analysis of predictive performance.** Description of creation of the synthetic data sets for predictability analysis.
(PDF)

**S4 Text. Compound screen data acquisition.** Description of the experimental protocol of the compound screens and data normalization.
(PDF)

**S5 Text. siRNA-knockdown of predicted host factors for HCV.** Description of the experimental protocol of the siRNA validation screen, data normalization and the statistical analysis.
(PDF)

**S6 Text. Details on estimation in linear mixed models.**
(PDF)

**S1 Data. Screening data.** Pharmacological inhibition screen, validation screen of network hits for HCV, and other relevant data to reproduce results and figures.
(ZIP)

**S1 Fig. Normalisation effect on readouts of plate controls.** Comparison of plate readouts for positive and negative controls.
(PDF)

**S2 Fig. Normalisation effect on control distributions.** Comparison between unnormalized and normalized control densities.
(PDF)

**S3 Fig. Pathogen-specific gene hits.** Visualization of the top pathogen-specific effect sizes.
(PDF)

**S4 Fig. Predictive performance on simulated and biological data.** The plots show the performance of our hierarchical model against PMM.
(PDF)

**S5 Fig. Identification of host factors using a random effects model.** The 25 first hits identified using the first step of our model are shown when sorting the estimates by absolute effect sizes.
(PDF)

**S6 Fig. Validation of identified host factors using pharmacological inhibition.** Description and visualization of the analysis of the validation screen using pharmacological inhibition.
(PDF)

## Acknowledgments

We thank Eric Snijder and Clara Posthuma for discussions and feedback regarding the siRNA screens. Moreover, we thank David Seifert and Christos Dimitrakopoulos for feedback on modelling the data set.

## Author Contributions

**Conceptualization:** Martijn van Hemert, Frank van Kuppeveld, Ralf Bartenschlager, Lars Kaderali, Marco Binder, Niko Beerenwinkel.

**Data curation:** Simon Dirmeier, Christopher Dächert, Martijn van Hemert, Ali Tas, Natacha S. Ogando, Frank van Kuppeveld, Ralf Bartenschlager.

**Formal analysis:** Simon Dirmeier.

**Funding acquisition:** Lars Kaderali, Marco Binder.

**Investigation:** Simon Dirmeier, Christopher Dächert, Martijn van Hemert, Ali Tas, Natacha S. Ogando, Frank van Kuppeveld, Ralf Bartenschlager, Lars Kaderali, Marco Binder.

**Methodology:** Simon Dirmeier.

**Project administration:** Lars Kaderali, Marco Binder, Niko Beerenwinkel.

**Resources:** Christopher Dächert, Martijn van Hemert, Frank van Kuppeveld, Ralf Bartenschlager.

**Software:** Simon Dirmeier.

**Supervision:** Lars Kaderali, Marco Binder, Niko Beerenwinkel.

**Validation:** Simon Dirmeier, Christopher Dächert.

**Visualization:** Simon Dirmeier.

**Writing – original draft:** Simon Dirmeier.

**Writing – review & editing:** Christopher Dächert, Martijn van Hemert, Ali Tas, Natacha S. Ogando, Frank van Kuppeveld, Ralf Bartenschlager, Lars Kaderali, Marco Binder, Niko Beerenwinkel.

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
