## [Decision Letter · Decision Letter 0]

9 Oct 2019

Dear Dr Beerenwinkel,

Thank you very much for submitting your manuscript 'Host factor prioritization for pan-viral genetic perturbation screens using random intercept models and network propagation' for review by PLOS Computational Biology. Your manuscript has been fully evaluated by the PLOS Computational Biology editorial team and in this case also by independent peer reviewers. The reviewers appreciated the attention to an important problem, but raised some substantial concerns about the manuscript as it currently stands. While your manuscript cannot be accepted in its present form, we are willing to consider a revised version in which the issues raised by the reviewers have been adequately addressed. We cannot, of course, promise publication at that time.

Sincerely,

Natalia L. Komarova

Deputy Editor

PLOS Computational Biology

Natalia Komarova

Deputy Editor

PLOS Computational Biology

[LINK]

Apologies for a long delay, I had a very difficult time finding both an AE and reviewers.

Reviewer's Responses to Questions

**Comments to the Authors:**

Reviewer #1: In this work, the authors have proposed a two-stage method that combines a statistical model and network propagation to prioritize host factors from heterogeneous and noisy RNAi screens for four different viruses. First a statistical model based on random effects model is used to rank the genes by their absolute effect size. In the second stage, a network propagation approach based on protein-protein interactions has been employed to study the effect of these genes and further fine tuning of list of prioritized genes that have significant impact on pan-viral life-cycle. The manuscript has been written nicely and the different methods and discussions are well-understandable. Although, there is not much computational novelty, still the computational work is supported with biological validation which is the key contribution of it. I have following comments on the manuscript.

1. The authors have demanded that their work is on pan-pathogen level. However they have used data for four viruses in this work. Is there any particular reason of choosing the four viruses? Do the authors expect similar results for other virus or pathogen combinations?

2. While describing the random effects model, the authors have used mathematical notations. It is understood that they have used some R package for this work, however, very limited description is there on how this is actually implemented, how the data sets are organized, or how the data set is fit into the model. These descriptions will beneficial for the readers.

3. An important stage of this work is the use of functional protein-protein interaction network in the second stage. However, there is practically no description of the corresponding data. The authors have just referred to the publication (ref. [39]) from which they have collected the data. However a short description is required for the sake of complete understanding. Another concern is that the data set is pretty old (2010). They should have used more recent interaction data instead.\\

4. There are few typos such as follows:

-- Line 230: The models described by Equation (1) and Equation (3) estimates gene effects  ... estimate ...

-- Use either 'knockdown' or 'knock-down' consistently.

Reviewer #2: The paper by Dirmeier et al. develops computational/statistical methods to identify host factors that are required by viruses for their replication, and that can be targets of therapeutic interference. While this approach has been done in select settings, such approaches to identify host factors on a viral group level are lacking. Identifying common host factors among a viral group would be desirable because this could allow the development of antivirals with broad-spectrum activity. A maximum likelihood approach using a random effects model was employed, and this information was propagated over a biological graph using network diffusion with Markov random walks. This method was able to reproduce previous work that identified host factors for single viruses and was then used to predict host factors across pathogen groups. This was validated by inhibition experiments.

This is an important topic and analysis, and the methodology seems well-developed and sound. I have some relatively minor comments:

- Statistical approaches to identify important host factors have been used in other settings, as outlined in the introduction of the paper (e.g. for single viruses, two viruses of the same genus, etc). Have these approaches been successful in developing treatments? If so, this could be summarized in the introduction. If not, this could be reviewed, and relevance for developing treatments as a result of the novel methodology could be discussed. Such additions would be useful for a more general computational biology readership.

- When identifying host factors that are relevant for a broad group of viruses, how do those factors that you identified compare with the factors that were identified in previous work in the context of more restricted settings? If a host factor is important for a broader group of viruses, is it likely that those host factors are more crucial for host cell function? If so, would that pose a problem to target them therapeutically? Perhaps a discussion of this could be added.

- The paper reports that no host factor could be found that is significant for all viruses in the group under consideration (+ ssRNA viruses). It could be useful to discuss the implications of this. Is it likely a general result that holds if you look at a different group of viruses? What are the limitations in the breadth of the viruses that can be treated by targeting a given host factor?

**Have all data underlying the figures and results presented in the manuscript been provided?**

Reviewer #1: Yes

Reviewer #2: Yes

PLOS authors have the option to publish the peer review history of their article (what does this mean?). If published, this will include your full peer review and any attached files.

Reviewer #1: No

Reviewer #2: No

---

## [Decision Letter · Decision Letter 1]

5 Dec 2019

Dear Dr Beerenwinkel,

We are pleased to inform you that your manuscript 'Host factor prioritization for pan-viral genetic perturbation screens using random intercept models and network propagation' has been provisionally accepted for publication in PLOS Computational Biology.

In the meantime, please log into Editorial Manager at https://www.editorialmanager.com/pcompbiol/, click the "Update My Information" link at the top of the page, and update your user information to ensure an efficient production and billing process.

One of the goals of PLOS is to make science accessible to educators and the public. PLOS staff issue occasional press releases and make early versions of PLOS Computational Biology articles available to science writers and journalists. PLOS staff also collaborate with Communication and Public Information Offices and would be happy to work with the relevant people at your institution or funding agency. If your institution or funding agency is interested in promoting your findings, please ask them to coordinate their releases with PLOS (contact ploscompbiol@plos.org).

Thank you again for supporting Open Access publishing. We look forward to publishing your paper in PLOS Computational Biology.

Sincerely,

Natalia L. Komarova

Deputy Editor

PLOS Computational Biology

Natalia Komarova

Deputy Editor

PLOS Computational Biology

Reviewer's Responses to Questions

**Comments to the Authors:**

Reviewer #1: The authors have answered all my inquiries satisfactorily in detail and also updated their manuscript. I believe this would be a valuable contribution to viral-host interaction research. I recommend publication of the manuscript in its current form.

**Have all data underlying the figures and results presented in the manuscript been provided?**

Reviewer #1: Yes

PLOS authors have the option to publish the peer review history of their article (what does this mean?). If published, this will include your full peer review and any attached files.

Reviewer #1: No

---

## [Editor Report · Acceptance letter]

3 Feb 2020

PCOMPBIOL-D-19-01066R1 

Host factor prioritization for pan-viral genetic perturbation screens using random intercept models and network propagation

Dear Dr Beerenwinkel,

I am pleased to inform you that your manuscript has been formally accepted for publication in PLOS Computational Biology. Your manuscript is now with our production department and you will be notified of the publication date in due course.

With kind regards,

Laura Mallard
